# Multimodal Imaging Assessment of Vascular and Neurodegenerative Retinal Alterations in Type 1 Diabetic Patients without Fundoscopic Signs of Diabetic Retinopathy

**DOI:** 10.3390/jcm8091409

**Published:** 2019-09-08

**Authors:** Riccardo Sacconi, Marco Casaluci, Enrico Borrelli, Giacomo Mulinacci, Francesca Lamanna, Francesco Gelormini, Adriano Carnevali, Lea Querques, Gianpaolo Zerbini, Francesco Bandello, Giuseppe Querques

**Affiliations:** 1Department of Ophthalmology, University Vita-Salute, IRCCS Ospedale San Raffaele, Via Olgettina 60, 20132 Milan, Italy (R.S.) (M.C.) (E.B.) (F.L.) (F.G.) (A.C.) (L.Q.) (F.B.); 2Complications of Diabetes Unit, Division of Metabolic and Cardiovascular Sciences, San Raffaele Scientific Institute, 20132 Milan, Italy (G.M.) (G.Z.)

**Keywords:** diabetic retinopathy, diabetes, dynamic vessel analyzer, microperimetry, OCTA, optical coherence tomography

## Abstract

The aim of this cross-sectional case-control study is to investigate the possible presence of vascular/neurodegenerative alterations in the retina of type 1 diabetes mellitus (T1DM) patients without diabetic retinopathy (DR). Thirty-four eyes of 34 consecutive T1DM without DR (mean age 21 ± 2 years) were included. Another cohort of 27 eyes (27 healthy control subjects matched with age and sex) was also recruited. All patients underwent multimodal imaging evaluation using structural optical coherence tomography (OCT), OCT-angiography (OCT-A), dynamic vessel analyzer (DVA) and microperimetry. No significant differences were disclosed comparing diabetics and controls for visual acuity, central macular thickness, and subfoveal choroidal thickness. On retinal nerve fiber layer and ganglion cell complex thickness, no significant differences were disclosed comparing each 3-mm-diameter macular and peripapillary subfield between two groups. Using OCT-A, deep capillary plexus perfusion density (PD) of diabetics was significantly lower compared to control group, whereas PD of other retinal/choriocapillaris plexuses and foveal avascular zone area did not show any significant difference. Using DVA, diabetic eyes revealed a significantly decreased vessel response to flicker light in comparison to controls. No differences were disclosed using microperimetry analysis. Taken together, these results suggest that vascular alterations could be the first detectable retinal change in the development of DR.

## 1. Introduction

Diabetic retinopathy (DR) is a leading cause of visual impairment in the working-age population worldwide [1]. Duration of diabetes is a major risk factor for the development and progression of the retinal disease [2,3]. The prevalence of DR is estimated to be around 10% in diabetic patients suffering for a period of five years, whereas retinopathy develops in nearly all diabetic patients suffering over 30 years [4]. Early detection of retinal alterations in diabetic patients and prompt treatment are key factors for the prevention of vision loss [5].

The introduction in the clinical practice of new diagnostic tools (i.e., high-resolution structural optical coherence tomography (OCT), and OCT-angiography (OCT-A)) has allowed us to identify pre-clinical retinal structural and functional alterations in patients without clinically funduscopic detectable signs of DR (i.e., microaneurysms formation, intraretinal hemorrhages, and/or microvascular changes) [6,7,8,9,10,11]. However, to date, there is still controversy about the exact mechanisms that lie beneath the development of DR in both type 1 and type 2 diabetes. Several studies have identified alterations in the inner retinal layers of patients without or with minimal DR using structural OCT, suggesting that detectable damage arising from diabetic neuroretinopathy could precede microvascular changes [6,12,13]. On the other hand, other authors reported microvascular changes using OCT-A in patients without DR [6,7,8,9,14,15]. OCT-A represents a relatively new noninvasive way to investigate retinal and choroidal vasculature in details. OCT-A has proved to be effective in the evaluation of qualitative features, such as nonperfusion areas, and also of quantitative metrics, including vessel density and area of foveal avascular zone (FAZ) [16,17,18,19]. In a previous study, we reported a significantly decreased vessel density in eyes of patients with type 1 diabetes mellitus (T1DM) compared to control eyes at the level of DCP using OCT-A [8].

Current technologies allow studying not only anatomical features but also functional retinal abnormalities in diabetic patients identified as without retinopathy at the fundoscopic examination. At this purpose, microperimetry offers the option to test retinal sensitivity while fundus is directly observed providing early macular function loss to be early detected before the occurrence of significant visual impairment [20]. Furthermore, dynamic vessel analyzer (DVA) represents a new method to study retinal blood vessels in a non-invasive way and allows the identification of early endothelial dysfunction [9,21].

In this study we integrated different retinal imaging technologies, namely structural OCT, OCT-A, DVA and microperimetry, in order to analyze both anatomical and functional features of the retina in a post-pediatric group of T1DM patients without fundoscopic signs of DR. Our purpose is to investigate the possible presence of vascular or neurodegenerative alterations in the retina of these young patients in order to better understand the mechanisms that lie beneath DR pathogenesis. The identification of such alterations during the subclinical phase of the disease could provide timely recognition of patients at a greater risk of DR progression and a better understanding of the pathogenesis of DR.

## 2. Methods

In this cross-sectional observational case-control study, consecutive patients with a diagnosis of T1DM without diabetic retinopathy were recruited at the Department of Ophthalmology of San Raffaele Hospital in Milan between May 2017 and April 2018 from a pool of patients previously followed at Pediatric Department. The study adhered to the 1964 Helsinki declaration and its later amendments. All included patients signed an informed consent that was approved by the Local Institutional Review Board (IRB of San Raffaele Hospital, Milan, Italy).

Enrollment criteria included: (1) age ≥ 18 years, (2) diagnosis of T1DM made, (3) at least five years of disease before enrollment, and (4) absence of any signs of DR at fundus biomicroscopy.

Exclusion criteria were: (1) history of any other retinal disease or previous intraocular intervention in the study eye, (2) myopia greater than six diopters (D) of sphere or three-dimensional (3D) cylinder, and/or axial length >25.5 mm, (3) insufficient clear ocular media or inadequate pupillary dilation or fixation that could affect quality imaging acquisition, history of any other retinal disease or previous intraocular intervention in the study eye.

If both eyes of a patient were eligible, only one eye was randomly chosen and included.

Another cohort of healthy control subjects matched with age and sex was also recruited. All healthy subjects had no ocular disorders and were visited by the senior author (GQ) in the Department of Ophthalmology, University Vita-Salute, San Raffaele Hospital in Milan.

All patients, both diabetics and controls, underwent a complete ophthalmologic evaluation including assessment of best-corrected visual acuity (BCVA) using Early Treatment Diabetic Retinopathy Study (ETDRS) charts, dilated slit-lamp anterior segment and fundus biomicroscopy. In addition, each enrolled patient underwent a multimodal imaging evaluation by means of structural spectral domain-OCT (SD-OCT), OCT-A, DVA, and microperimetry.

### 2.1. Structural SD-OCT Measurements

Structural SD-OCT images were acquired using Spectralis (Heidelberg Engineering, Heidelberg, Germany). Central macular thickness (CMT) in the central 1-mm-diameter circle of the ETDRS thickness map was provided by the Spectralis software (Heidelberg Eye Explorer, version 1.9.11.0 Heidelberg Engineering, Germany) and recorded. Automated macular and peripapillary retinal nerve fiber layer (RNFL), macular ganglion cell complex (GCC), outer plexiform layer (OPL), and outer nuclear layer (ONL) thickness were also recorded using the abovementioned inbuilt software. In detail, 1-mm-diameter central circle (C) quadrant and nasal (N), temporal (T), superior (S), and inferior (I) quadrants of 3-mm-diameter and 6-mm-diameter subfield as defined by ETDRS were recorded for macular thickness analysis. With regard to peripapillary RNFL, central (G), nasal (N), nasal superior (NS), temporal superior (TS), temporal (T), temporal inferior (TI), and nasal inferior (NI) quadrants of 4.1-mm-diameter subfield were recorded and used for the analysis. Furthermore, as Spectralis OCT does not provide an automatic segmentation of the ellipsoid zone (EZ), outer segment (OS) surface, and choroid, we manually measured the subfoveal EZ thickness (EZ was defined as the reflective layer situated posterior to the weak-reflecting ONL and anterior to the strong-reflecting retinal pigment epithelium (RPE)), OS thickness (OS was defined as the weak-reflecting layer between the EZ and RPE), and choroid thickness (ChT; ChT was defined as the distance between Bruch’s membrane interface and the sclerochoroidal interface). All values were manually measured by two expert readers (RS and EB) and the mean value was used for statistical analysis.

### 2.2. OCT-A Image Acquisition and Analysis

OCT-A examinations were performed using Swept-Source OCT-A PLEX^®^ Elite 9000 (Carl Zeiss Meditec, Inc., Dublin, CA, USA). A scanning area of 3 × 3 mm centered on the foveal area was adopted for all patients. FastTrac motion correction software was used during images acquisition to reduce motion artifacts. In order to calculate the FAZ area and Perfusion Density (PD), a Mean’s thresholding and binarization process were made for each 3 × 3 OCT-A image by using ImageJ image processing program, according to the previous studies [9,22,23]. In detail, FAZ area was manually marked out using the polygon selection tool in superficial capillary plexus (SCP) and deep capillary plexus (DCP) and colored to pure blue and its dimension was expressed as square millimeters (mm^2^). PD was calculated for superficial capillary plexus (SCP), deep capillary plexus (DCP) and choriocapillaris plexus (CCP) as the ratio between the white pixels and the total pixels after the FAZ had been excluded.

### 2.3. Dynamic Vessel Analysis

All DVA examinations (Imedos, Jena, Germany) were performed by a single trained grader (MC) at approximately the same time in the afternoon (between 2 pm and 4 pm). All subjects were required to stay off alcohol and caffeine-containing products in the 24 h prior to the study.

Pupils were dilated with tropicamide eyedrops and the analysis was performed in a dimly lit room. To perform the dynamic analysis in type 1 diabetic patients and in healthy controls, a superior or inferior temporal venous and arterial segment located between one half and two-disc diameters from the optic disc margin were chosen. The selected vessel was at least one vessel diameter from any bifurcation or closed vessel, and it was marked manually with a probe (blue for the vein and red for the artery).

During the examination (350 s), three cycles of flicker/nonflicker light were registered, as reported in previous studies [24,25,26,27]. The selected vessel diameters were first recorded for 50 s before flicker stimulation and for 80 s after the flickering light. Vessel diameters were expressed in measurement units (MU); vessel dilation was measured by calculating the percentage increase in vessel diameter relative to baseline after 20 s of flicker stimulation and averaging the three measurement cycles.

### 2.4. Static Vessel Analysis

The retinal vessel analyzer (RVA; Imedos GmbH, Jena, Germany) is a commercially available tool for the assessment of retinal vessel diameter in relation to time. Using the FF450 retinal camera (Carl Zeiss GmbH, Jena, Germany), included in the DVA system, a 50-degree fundus photograph was acquired in each subject. VISUALIS and VesselMap Software version 3.10 (Imedos Systems, Ltd., Jena, Germany) allowed the analysis of these photographs. In all subjects, we calculated the central retinal artery equivalent (CRAE), which relates to the diameter of the central retinal artery and the central retinal vein equivalent (CRVE), which relates to the diameter of the central retinal vein.

### 2.5. Microperimetry Assessment

Assessment of macular sensitivity was performed with the MP-1 microperimeter (Nidek Technologies, Padova, Italy). Before starting the test all participants underwent a mesopic adaptation period of 15 min as microperimetry needs the eye to work on mesopic range in order to measure retinal sensitivity appropriately. The same settings were applied for all examinations: a customized grid of 33 Goldmann II stimuli, covering the central 10° (centered on the fovea), was created. Stimuli were presented in random order according to a 4-2-1 double-staircase strategy to generate retinal sensitivity maps. The stimulus intensity ranged from 0 dB, corresponding to the strongest signal intensity of 127 cd/m^2^, to 20 dB in 1-dB steps, and the duration of each stimulus was 200 milliseconds.

### 2.6. Statistical Analysis

Statistical evaluation was performed using SPSS statistics software version 22.0 (SPSS Inc., IBM, Chicago, IL, USA). All quantitative data were expressed as mean ± standard deviation (SD). Continuous variables were tested for normal distributions, according to the Kolmogorov-Smirnov test. Chi-squared test was used to analyze categorical variables. Comparisons of mean values of quantitative variables between diabetic patients and control subjects were made using a Student’s independent *t*-test. Correlation between continuous variables was assessed calculating Pearson’s correlation coefficients. A *p* value ≤ 0.05 was considered to be statistically significant.

## 3. Results

### 3.1. Patients Demographics and Main Clinical Findings

A total of 34 eyes of 34 consecutive T1DM patients (16 females, 18 males) without signs of DR were included in the study. The mean age was 21 ± 2 years (median 21; range 18–25 years), and all patients were Caucasian. Mean duration of the disease was 12 ± 4 years (range 5–20 years), and mean HbA1c level was 7.6 ± 0.7% (range 6.0–9.1%). None of the patients was affected by systemic hypertension and renal dysfunction. BCVA was 20/20 Snellen equivalent in all eyes. Mean CMT was 277 ± 16 µm (range 243–312 µm), and mean subfoveal ChT was 299 ± 62 µm (range 204–466 µm).

Thirty-two healthy subjects were included in the control group. Patients included were homogenous for age and sex with the diabetic group: the mean age was 22 ± 2 years (median 23; range 18–25 years) (*p* = 0.141), with 16 females and 16 males (*p* = 0.811). No significant differences were disclosed comparing diabetics and controls for BCVA, CMT, and subfoveal ChT. In detail, control group BCVA was 20/20 Snellen equivalent in all eyes (*p* = 1.000), CMT was 273 ± 17 µm (range 246–314 µm; *p* = 0.285), and subfoveal ChT was 280 ± 81 µm (range 185–460 µm; *p* = 0.433). The main clinical findings are summarized in Table 1.

### 3.2. Structural OCT Analysis

In order to evaluate structural inner and outer retinal changes, we analyzed the thickness of RNFL, GCC, OPL, and ONL in the 3-mm and 6-mm-diameter macular subfields. On macular RNFL, GCC, OPL, and ONL thickness analysis, no significant differences were disclosed comparing each 3-mm-diameter subfield (C, S, I, N and T) of diabetic patients with the corresponding subfield of control subjects, whereas with regard to GCC 6-mm-diameter subfield diabetic patients revealed a significant increased thickness in different quadrants compared to controls (Table 2 and Table 3). Furthermore, we analyzed the subfoveal EZ and OS thickness and we did not disclose any significant difference between diabetic patients and controls (EZ: 20.5 ± 2.5 µm and 21.1 ± 2.8 µm, respectively (*p* = 0.355); OS: 34.8 ± 3.7 µm and 34.2 ± 3.0 µm, respectively (*p* = 0.497)).

In addition, no differences were disclosed comparing each quadrant (G, N, NS, TS, T, TI, and NI) of peripapillary RNFL of the two groups. All values of analysis were reported in Table 2.

### 3.3. OCT-A Analysis

OCT-A analysis was performed to detect early changes in retinal and CC vascular plexuses. FAZ area was well-detectable in all diabetic and control eyes. No significant difference was disclosed in the FAZ area at both SCP and DCP by comparing diabetics and control group (Figure 1). Mean FAZ area in SCP was 0.235 ± 0.072 mm^2^ in diabetic eyes and 0.199 ± 0.100 mm^2^ in control eyes (*p* = 0.122). In DCP, FAZ area was 0.670 ± 0.178 mm^2^ and 0.620 ± 0.257 mm^2^ (diabetic and healthy subjects, respectively (*p* = 0.391)).

With regard to PD analysis of the retinal and choriocapillaris vessels, no difference was disclosed in SCP and CC between patients affected by diabetes and control subjects (SCP: 0.470 ± 0.009 versus 0.466 ± 0.010 for diabetic and healthy patients, respectively (*p* = 0.209); CC: 0.506 ± 0.014 and 0.503 ± 0.010 for diabetic and healthy patients, respectively (*p =* 0.386)). On the other hand, at the level of DCP, diabetic eyes revealed a significantly decreased PD compared to the control group (0.449 ± 0.013 versus 0.458 ± 0.012, respectively (*p =* 0.013)] (Figure 1).

Sex, patient age, HbA1c level, and duration of the disease did not influence significantly the FAZ area and PD in SCP, DCP, and CCP (*p* > 0.1 in all analyses).

### 3.4. Dynamic Vessel Analysis

In order to evaluate the response of retinal vessels to flickering light using DVA, we measured the vessel dilation (arterial and venous) as the percentage increase in vessel diameter relative to baseline after 20 s of flicker stimulation. Diabetic eyes revealed a significantly decreased vessel response to flicker light. Of note, mean arterial dilation percentage of diabetic patients and controls was 1.7 ± 2.2% and 3.8 ± 2.0%, respectively (*p* < 0.001). Vein dilation percentage was 2.6 ± 2.3% for diabetic patients and 4.1 ± 1.6% for healthy subjects (*p* = 0.005) (Figure 2).

### 3.5. Static Vessel Analysis

The static analysis disclosed no significant difference in mean CRAE and mean CRVE between the two groups. Since CRAE and CRVE relate to the diameter of the central retinal artery and vein, respectively, we did not disclose any significant difference in the mean diameter of major retinal vessels between diabetic patients and controls. In detail, the mean CRAE of diabetic patients was 239.7 ± 54.3 MU, whereas the mean CRAE of controls was 231.7 ± 40.5 MU (*p* = 0.527). The mean CRVE was 254.2 ± 39.3 and 255.0 ± 26.9 MU in diabetic and controls, respectively (*p* = 0.927) (Figure 2).

### 3.6. Microperimetry Analysis

In order to evaluate functional alterations, all patients underwent microperimetry after mesopic adaptation period of 15 min. No differences were disclosed in retinal sensitivity by microperimetry (Figure 3). Mean retinal sensitivity (MS) in the two groups analyzed was 13.4 ± 2.1 dB and 13.9 ± 2.2 dB, respectively (*p* = 0.347). In order to evaluate the retinal sensitivity of the central and paracentral area, we analyzed the MS of 13 Goldmann II stimuli inside the central 4° (central area) and the MS of 20 Goldmann II stimuli between the 4° and the 10° (paracentral area). In detail, no significant differences were disclosed in both areas between diabetic patients and controls. MS of central area was 15.0 ± 1.9 dB and 15.0 ± 2.1 dB for diabetic patients and controls, respectively (*p* = 0.962). MS of paracentral area was 12.5 ± 2.2 dB and 13.2 ± 2.4 dB for diabetic patients and controls, respectively (*p* = 0.251). Mean fixation percentage calculated within the central 2° (centered on the fovea) was 95.2 ± 5.6% in diabetic patients and 94.7 ± 4.6% in controls (*p* = 0.724), whereas mean fixation percentage within the central 4° (centered on the fovea) was 98.9 ± 1.8% and 98.2 ± 2.5% in diabetic patients and controls, respectively (*p* = 0.225).

## 4. Discussion

In this study, we performed a multimodal imaging evaluation to analyze possible early structural and/or functional alterations in post-pediatric patients with T1DM without DR. Using OCT-A, we disclosed a significant lower PD in the DCP in diabetic eyes compared to control eyes, whereas no significant difference between two groups was disclosed with regard to PD of SCP and CCP and with regard to the FAZ area (Figure 1). Furthermore, our results showed a significantly decreased vessel response to flicker light stimulation occurring in T1DM patients with no signs of DR compared with healthy subjects using DVA (Figure 2). On the other hand, no significant changes were disclosed in the central RNFL and GCC analysis, and in the retinal sensitivity using microperimetry (Figure 3).

According to these results, a decreased PD of the DCP could be considered one of the earliest detectable signs of microvascular sufferance in the retina of type 1 diabetic patients, although probably not sufficient to alter the CMT significantly. Moreover, no impairment in the choroid circulation seems to characterize the earlier stages of retinopathy.

In a recent study, Cao et al. [9] analyzed the eyes of 71 patients with type 2 diabetes mellitus (T2DM) with no signs of DR and 67 control subjects, and disclosed that T2DM patients had a reduced vessel density in SCP, DCP and choriocapillaris compared to healthy subjects, suggesting an involvement of both retinal and choroidal circulation before clinical manifestation of DR. Dimitrova et al. [15] reported a reduction in parafoveal superficial and deep retinal vessel density in diabetic patients compared to controls. However, no difference between type 1 and type 2 DM was made in this case. Similar results were reported also by Zeng et al. in a cohort of T2DM patients [28]. As T1DM patients with no signs of DR are on average younger than type 2 DM patients and usually have a lower prevalence of comorbidities that may contribute to retinal vascular changes (i.e., hypertension and dyslipidemia), a T1DM-only cohort may be better suited to study retinal changes due specifically to the metabolic dysregulation of diabetes [29]. In this regard, our group previously analyzed the retinal microvasculature of young T1DM patients without any signs of DR by means of SD-OCT-A and disclosed that vessel density was reduced in the DCP, but not in SCP and CCP compared to controls [8]. Similar results, in a cohort of only T1DM patients with no or mild signs of retinopathy, were reported also by Simonett et al. [7]. There are many reasons why the DCP could be impaired by the metabolic complications of diabetes at an earlier stage compared to SCP and CCP. The most relevant one could be due to the complex vascular anatomical organization of DCP, which involves distance from the larger superficial arterioles and a terminal-type capillary bed. Probably, the involvement of the other vascular plexuses highlighted by other studies could reflect a more advanced stage of the disease as the mean age of patients enrolled was higher and the duration of diabetes was longer in most cases.

Recently, controversial results about early alterations in FAZ area have been evaluated by means of OCT-A in diabetic eyes without signs of retinopathy. FAZ size was disclosed to be significantly increased in diabetic patients compared to control by De Carlo et al. [30], whereas no significant difference was disclosed in FAZ area of both SCP and DCP by Scarinci et al. [31]. In our current study, no FAZ enlargement has been revealed in diabetic eyes, suggesting that alterations in FAZ area can occur subsequently to dropout of perifoveal capillaries causing defects in PD. Furthermore, due to the high variability that characterizes FAZ area in healthy subjects, this should probably not be considered a good biomarker of the earlier stages of retinopathy

At present, it is not well understood whether the earliest abnormalities in diabetic patients involve retinal microcirculatory bed or retinal inner neural layers. Previously, some authors reported early damage involving RNFL and GCC in T1DM subjects with no signs of DR, thus proposing that neuroretinal degeneration occurs prior to the development of vascular changes [6,28,32]. Nevertheless, it must be kept in mind that the abovementioned studies labeled patients as not having retinopathy simply on the basis of fundus examination. However, thanks to the introduction in clinical practice of OCT-A, we are able to recognize the earliest changes involving retinal microcirculation long before they can be visualized by fundus biomicroscopy. The results of the current study did not show any significant change in macular and peripapillary RNFL thickness, suggesting that a vascular alteration, in particular at the level of the DCP, probably precedes detectable neuroretinal impairment in T1DM patients. Furthermore, as for GCC and OPL thickness, no differences were disclosed analyzing 3-mm-diameter subfield quadrants between the two groups, whereas diabetic patients showed increased thickness in 6-mm-diameter subfields compared with controls. The meaning of this finding is not fully clear to us, but we hypothesize it could be the result of swelling due to intracellular fluid accumulation. Recent studies in animal models suggested that retinal photoreceptor cells have a crucial role in the pathogenesis of retinal microvascular lesion in diabetes [33]. However, our in vivo analysis did not show any significant changes in foveal OPL, ONL, EZ, and OS surface thickness using structural OCT but a reduced PD of the DCP using OCT-A. Nevertheless, we cannot completely exclude the possibility that early photoreceptors changes, not detectable using structural OCT technology, could be the primary trigger in the DR development. Indeed, we tried to disclose neurodegenerative and photoreceptor changes analyzing anatomical changes on structural OCT scans. However, this technology is not able to detect intracellular anatomical changes (not affecting the retinal thickness) and/or functional cellular changes.

To better understand the physiopathological mechanisms that lie beneath the development of retinal alterations in diabetic patients, we also performed DVA and microperimetry in our study. While by means of OCT and OCT-A we can get important information on the retina by a structural point of view, DVA and microperimetry allow us to test the retina by a functional point of view. Previously, Lim et al. [34] investigated the responses of retinal vessels to flickering light in diabetic patients with various stages of diabetic retinopathy and reported that the responses of retinal arterioles and venules to flickering light decrease progressively from earlier to more severe stages of DR. However, they did not include in their study diabetic patients without diabetic retinopathy. Our results showed a significantly decreased vessel response to flicker light stimulation occurring in T1DM patients with no signs of DR compared with healthy subjects. Although these findings suggest an early vascular dysfunction in diabetic patients, it must be considered that the functional hyperemia response to flickering light is actually mediated by neurovascular coupling [35,36,37]. In fact, flickering light can activate amacrine cells and ganglion cells in the inner retina [38] and induces the release of vasodilating factors not only from endothelial cells but also from neural cells. Hence, an altered vessels response to flickering light may be the result of vascular as well as neurodegenerative alterations.

Using micromerimetry, no difference was disclosed by comparing retinal sensitivity in diabetics and control subjects. Since microperimetry evaluates subjective retinal functional abnormalities and provides early detection of retinal pathology [20], our results suggest that neurodegenerative processes may appear in a later stage of the disease compared to alterations involving microvasculature architecture or function as shown by means of OCT-A and DVA. Recently, Zeng et al. [28], using flicker electroretinography, reported a delayed implicit time and decreased amplitude in patients affected by T2DM without diabetic retinopathy, reporting an early functional change. However, the authors disclosed also anatomical changes, with reduced vessel density of retinal capillary plexuses and reduced RNFL thickness. Bearing in mind that the functional tests used in the study by Zeng et al. [28] are not completely comparable to those used in our current study, together with different patient populations between study, we would like to hypothesize that different results here disclosed could be related to analysis at an earlier stage in our T1DM patients in comparison to T2DM patients analyzed by Zeng et al. [28].

The limitations of this study should be kept in mind. First of all, this study incorporated a relatively small sample size. However, this is a very particular population, with only post-pediatric patients affected by T1DM with a mean duration of the disease of 12 years. Moreover, despite the evidence of our results, we cannot completely exclude that neurodegenerative alterations may be the primary trigger in the DR development. Early neurodegenerative alterations maybe not revealed by actual technology but may be highlighted in the future with the development of new imaging modalities. Indeed, structural OCT disclosed neurodegenerative changes analyzing the thickness of retinal layers. However, this technology is not able to detect intracellular anatomical changes (not affecting the retinal thickness) and/or functional cellular changes. Another limitation of our study is that we performed microperimetry in order to evaluate the retinal sensitivity, but other tests for retinal and visual function (i.e., visual field or dark adaptation examinations) could show different results. Furthermore, our study analyzed mainly the 6 × 6 mm macular area; however, more peripheral retina analysis (outside 6 mm diameter) could show earlier alterations in diabetic patients.

## 5. Conclusions

In conclusion, we reported early impairment in PD of the DCP and decreased vessel response to flicker light stimulation in T1DM without fundoscopic signs of DR. On the other hand, no significant changes were disclosed in RNFL and central GCC thickness, and using microperimetry. Taken together, these results suggest that vascular alterations could be the first detectable retinal change in the development of DR. Future long-term follow-up studies are warranted to better understand the evolution of the disease and subsequent steps in DR development.

## Figures and Tables

**Figure 1 jcm-08-01409-f001:**
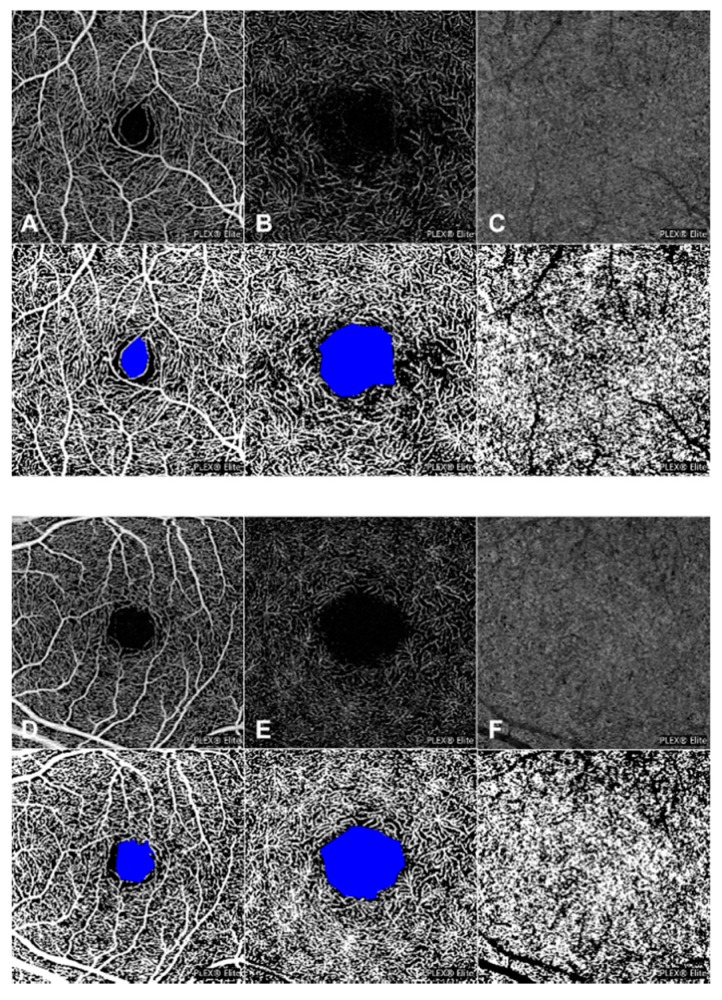
Optical coherence tomography angiography (OCT-A) analysis of a type 1 diabetic patient and a healthy control subject. 3 × 3 en-face OCT-A images with corresponding binarized images of superficial capillary plexus (SCP) (**A**), deep capillary plexus (DCP) (**B**), and choriocapillaris (CC) plexus (**C**) of a type 1 diabetic patient and of a healthy control subject (**D**–**F**). No significant differences were disclosed in the perfusion density (PD) of SCP and CC between diabetic patients (**A**,**C**) and controls (**D**,**F**), but diabetic eyes revealed a significantly decreased PD compared to the control group in the DCP (**B**,**E**). In the binarized image, FAZ area of SCP and DCP was colored with pure blue.

**Figure 2 jcm-08-01409-f002:**
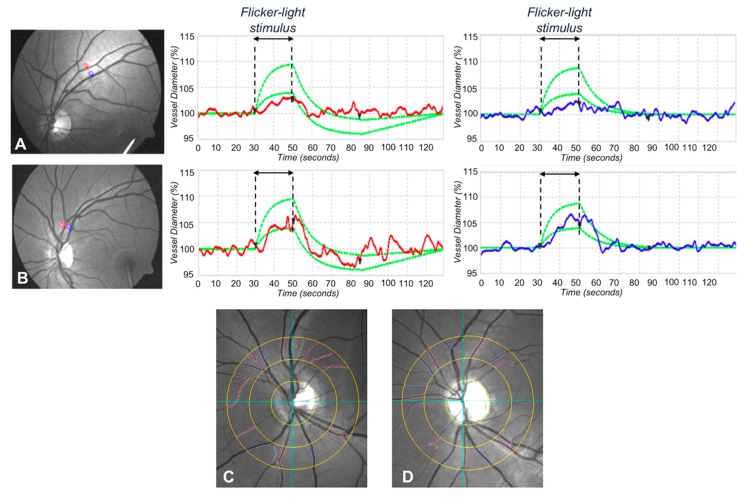
Dynamic and static vessel analysis of a patient with type 1 diabetes without diabetic retinopathy and of a healthy control subject. Dynamic vessel analysis of a diabetic patient. (**A**) Arterial and venous segments are chosen and marked with a probe (upper left panel, red for the artery, and blue for the vein) to evaluate the arterial (upper middle panel) and venous (upper right panel) flicker response. Similarly, in healthy control subjects (**B**), arterial and venous segments are chosen and marked with a probe (lower left panel, red for the artery and blue for the vein) to evaluate the arterial (lower middle panel) and venous (lower right panel) flicker response. In dynamic vessel analysis, diabetic eyes revealed a significantly decreased vessel response to flicker light in both arterial and venous dilation in comparison with controls (**A**,**B**). Static vessel analysis (**C**,**D**). Arterial and venous vessels are selected manually to calculate the central retinal artery equivalent and central retinal vein equivalent. The same procedure is repeated in type 1 diabetic patients (**C**) and in healthy control subjects (**D**). No significant difference in static vessel analysis was disclosed between the two groups.

**Figure 3 jcm-08-01409-f003:**
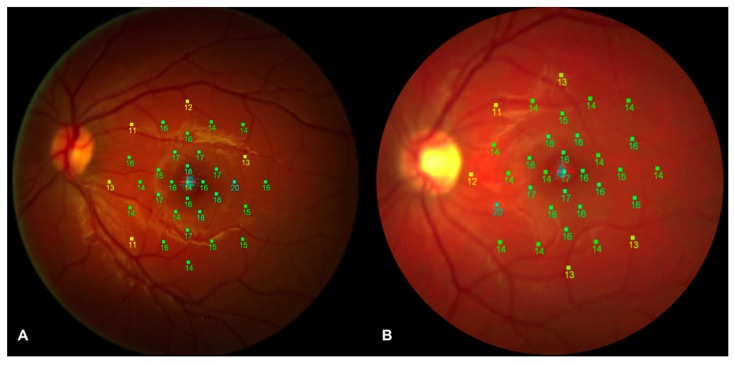
Microperimetry analysis of a diabetic patient and a healthy control subject. Retinal sensitivity map of a diabetic patient (**A**) and of a healthy control subject (**B**) registered along with a color fundus photograph. No significant difference in mean retinal sensitivity was disclosed between the two groups.

**Table 1 jcm-08-01409-t001:** Demographic and clinical characteristics of study population.

	Diabetic Eyes (*n* = 34)	Control Eyes (*n* = 32)	*p* Value
Gender (male/female)	18/16	16/16	0.811 *
DM duration (mean ± SD), years	12 ± 4	\	\
HbA1c, %	7.6 ± 0.7	\	\
Age (mean ± SD), years	21 ± 2	22 ± 2	0.141 ^+^
BCVA (mean ± SD), LogMAR	0 ± 0	0 ± 0	1.000 ^+^
CMT (mean ± SD), µm	277 ± 16	273 ± 17	0.285 ^+^
Subfoveal ChT (mean ± SD), µm	299 ± 62	280 ± 81	0.443 ^+^

*n*, number; SD, standard deviation; DM, diabetes mellitus; HbA1C, haemoglobin A1c; BCVA: best-corrected visual acuity; CMT: central macular thickness; ChT: choroidal Thickness. *: *χ*^2^ test; +: Student’s *t* test for independent samples.

**Table 2 jcm-08-01409-t002:** GCC and RNFL thickness analysis of diabetic eyes compared with control group.

Subfield Analyzed	Diabetic Eyes (*n* = 34)	Control Eyes (*n* = 32)	
	Mean ± SD	Mean ± SD	*p* Value *
GCC thickness (µm)			
1-mm central circle	16.1 ± 3.3	15.7 ± 3.3	0.653
3-mm S subfield	54.9 ± 5.8	54.6 ± 3.4	0.789
6-mm S subfield	37.5 ± 3.2	34.9 ± 3.1	0.001
3-mm I subfield	53.8 ± 5.2	53.6 ± 3.3	0.854
6-mm I subfield	37.2 ± 4.8	34.9 ± 3.2	0.025
3-mm N subfield	54.2 ± 5.0	52.9 ± 3.5	0.229
6-mm N subfield	40.0 ± 4.1	38.7 ± 3.3	0.167
3-mm T subfield	49.3 ± 6.0	49.9 ± 4.2	0.651
6-mm T subfield	39.6 ± 4.7	36.5 ± 3.5	0.004
Macular RNFL thickness (µm)			
1-mm central circle	12.7 ± 1.7	12.6 ± 1.3	0.832
3-mm S subfield	24.2 ± 3.1	23.6 ± 3.0	0.390
6-mm S subfield	36.9 ± 5.0	35.6 ± 4.4	0.262
3-mm I subfield	24.5 ± 2.4	24.1 ± 2.5	0.477
6-mm I subfield	39.9 ± 6.7	37.7 ± 5.3	0.130
3-mm N subfield	21.2 ± 1.9	20.7 ± 1.8	0.227
6-mm N subfield	50.5 ± 6.4	48.6 ± 6.8	0.240
3-mm T subfield	16.2 ± 0.9	16.0 ± 1.1	0.334
6-mm T subfield	17.7 ± 0.9	17.9 ± 1.2	0.332
Peripapillary RNFL thickness (µm)			
G	87.7 ± 8.0	88.0 ± 9.8	0.912
N subfield	67.7 ± 9.5	73.6 ± 15.1	0.234
NS subfield	102.4 ± 17.9	100.6 ± 22.8	0.720
TS subfield	116.3 ± 25.1	121.0 ± 19.1	0.415
T subfield	67.0 ± 9.9	66.2 ± 15.6	0.798
TI subfield	126.1 ± 26.0	132.3 ± 19.8	0.297
NI subfield	100.6 ± 24.9	93.6 ± 24.7	0.268

GCC: ganglion cell complex; RNFL: retinal nerve fiber layer; *n*: number; SD: standard deviation; S: superior; I: inferior; N: nasal; T: temporal; G: central. *: Student’s *t* test for independent samples.

**Table 3 jcm-08-01409-t003:** OPL and ONL thickness analysis of diabetic eyes compared with control group.

Subfield Analyzed	Diabetic Eyes (*n* = 34)	Control Eyes (*n* = 32)	
	Mean ± SD	Mean ± SD	*p* Value *
OPL thickness (µm)			
1-mm central circle	26.8 ± 4.9	25.9 ± 4.3	0.475
3-mm S subfield	38.0 ± 8.8	35.0 ± 9.9	0.179
6-mm S subfield	27.9 ± 2.8	26.0 ± 3.1	0.015
3-mm I subfield	31.5 ± 4.0	30.7 ± 6.5	0.599
6-mm I subfield	26.5 ± 1.7	24.9 ± 1.8	0.001
3-mm N subfield	32.8 ± 7.7	30.3 ± 4.7	0.133
6-mm N subfield	28.4 ± 2.9	26.4 ± 2.7	0.007
3-mm T subfield	33.8 ± 5.8	34.4 ± 7.0	0.713
6-mm T subfield	27.9 ± 2.1	26.9 ± 3.1	0.132
ONL thickness (µm)			
1-mm central circle	91.5 ± 9.3	88.7 ± 11.1	0.274
3-mm S subfield	67.8 ± 11.2	65.7 ± 12.0	0.474
6-mm S subfield	63.4 ± 7.4	61.2 ± 8.1	0.242
3-mm I subfield	72.7 ± 7.8	69.2 ± 9.0	0.100
6-mm I subfield	58.1 ± 7.4	55.4 ± 7.3	0.147
3-mm N subfield	75.1 ± 11.4	74.5 ± 11.2	0.834
6-mm N subfield	61.1 ± 7.5	59.5 ± 8.9	0.438
3-mm T subfield	71.5 ± 10.5	66.6 ± 10.3	0.061
6-mm T subfield	61.4 ± 7.9	56.8 ± 7.1	0.017

OPL: outer plexiform layer; ONL: outer nuclear layer; *n*: number; SD: standard deviation; S: superior; I: inferior; N: nasal; T: temporal; G: central. *: Student’s *t* test for independent samples.

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
