# Peer review of "Multimodal Imaging Assessment of Vascular and Neurodegenerative Retinal Alterations in Type 1 Diabetic Patients without Fundoscopic Signs of Diabetic Retinopathy"

_jcm, 2019, doi:10.3390/jcm8091409_

Round 1

Reviewer 1 Report

In this manuscript the authors are reporting the results of a study focusing on testing different technology assessing retinal anatomy and function in a population of diabetic patients without any clinical signs of diabetic retinopathy. This study is building in the relatively recent paradigm shift in the diabetic retinopathy field regarding the early impact of diabetes on the neuroretina and its function even in absence of clinical signs (vascular) of DR. None of the technologies or their use in diabetic patients are really novel but their assessment in diabetic patients without DR is nevertheless interesting and meaningful as it could lead to the identification of subpopulation either at risk or targetable for drug trials or “down the road” treatment. While overall this study has merit, there needs to be some significant improvement made.

Specific comments:

The major issue with this study, and it is partially eluded to by the authors in the discussion, is the relatively small sample size. While this reviewer completely understand the difficulty of recruitment for the population studied and acknowledge that the sample size is already decent, it becomes a really significant problem in the interpretation of the data of this study as there are several endpoints that are really close to the cut-off of 0.05 and thus leaves a lot of room for questioning how many of those are missed as actual changes due to an underpowered study. This would be most likely solved by adding only a few additional samples to each group, maybe even just the control group.

The second major criticism for this study is the absence of other recently identified more sensitive assays for retinal and visual function in addition to microperimetry (visual field, dark adaptation…).

The third major point is in regard to microperimetry and structural OCT analysis as those are too limited in size, with the OCT only going as far as 6mm and microperimetry is only analyzed as mean sensitivity when the one example given clearly suggests that there are more localized (relatively peripheral) changes.

Overall, it seems that the observation might be too central. Lots of studies suggest that central and peripheral retina changes might need to be analyzed, especially early on (hypoperfusion tends to be primarily in the periphery, and the dilation results of this specific study, which are showing significant changes are not that central). Again, this is also supported by the microperimetry data in this paper for which there seems to be reduction in the one example shown when focusing more peripherally.

Presentation/organization is confusing. It is unclear by the way that the file is formatted where the figure legends start and finish. In any case, this needs to be verified and clarified to make it so that the reader can follow easily the figures even independently from the result section (i.e. what does each letter in each figure mean?).

Overall, the manuscript is fairly well-written but there remains a few instances of word misuse (disclose instead of found/observed) or poor formulation (“diabetic eyes revealed” instead of “the analysis of diabetic eyes revealed”) that need to be corrected.

Author Response

In this manuscript the authors are reporting the results of a study focusing on testing different technology assessing retinal anatomy and function in a population of diabetic patients without any clinical signs of diabetic retinopathy. This study is building in the relatively recent paradigm shift in the diabetic retinopathy field regarding the early impact of diabetes on the neuroretina and its function even in absence of clinical signs (vascular) of DR. None of the technologies or their use in diabetic patients are really novel but their assessment in diabetic patients without DR is nevertheless interesting and meaningful as it could lead to the identification of subpopulation either at risk or targetable for drug trials or “down the road” treatment. While overall this study has merit, there needs to be some significant improvement made.

Thank you for your favorable comment.

Specific comments:

The major issue with this study, and it is partially eluded to by the authors in the discussion, is the relatively small sample size. While this reviewer completely understand the difficulty of recruitment for the population studied and acknowledge that the sample size is already decent, it becomes a really significant problem in the interpretation of the data of this study as there are several endpoints that are really close to the cut-off of 0.05 and thus leaves a lot of room for questioning how many of those are missed as actual changes due to an underpowered study. This would be most likely solved by adding only a few additional samples to each group, maybe even just the control group.

Thank you for your suggestion. As suggested by the reviewer, we added some subjects in the control group and we reperform the whole statistical analysis. As reported in the results’ section, previous results were confirmed by the new analysis. Now, no endpoints are close to the cut-off of 0.05. Please see all the revised results’ section of the manuscript.

The second major criticism for this study is the absence of other recently identified more sensitive assays for retinal and visual function in addition to microperimetry (visual field, dark adaptation…).

Thank you for your comment. We added this criticism in the revised version of the manuscript (page 10, line 799-801): “Another limitation of our study is that we performed the microperimetry in order to evaluate the retinal sensitivity, but other tests for retinal and visual function (e. visual field or dark adaptation examinations) could show different results”.

The third major point is in regard to microperimetry and structural OCT analysis as those are too limited in size, with the OCT only going as far as 6mm and microperimetry is only analyzed as mean sensitivity when the one example given clearly suggests that there are more localized (relatively peripheral) changes. Overall, it seems that the observation might be too central. Lots of studies suggest that central and peripheral retina changes might need to be analyzed, especially early on (hypoperfusion tends to be primarily in the periphery, and the dilation results of this specific study, which are showing significant changes are not that central). Again, this is also supported by the microperimetry data in this paper for which there seems to be reduction in the one example shown when focusing more peripherally.

Thank you very much for your suggestion. In order to evaluate the central area and the more peripheral area (i.e. paracentral area) of the microperimetry, we performed a subanalysis of the microperimetry in our patients. In detail, we analyzed the MS of 13 Goldmann II stimuli inside the central 4° (central area) and the MS of 20 Goldmann II stimuli between the 4° and the 10° (paracentral area). In this subanalysis, no reduction of the retinal sensitivity was disclosed between diabetic patients and controls in both central and paracentral area. We reported this new analysis in the revised version of the manuscript (page 7, line 622-646): “In order to evaluate the retinal sensitivity of the central and paracentral area, we analyzed the MS of 13 Goldmann II stimuli inside the central 4° (central area) and the MS of 20 Goldmann II stimuli between the 4° and the 10° (paracentral area). In detail, no significant differences were disclosed in both areas between diabetic patients and controls. MS of central area was 15.0±1.9 dB and 15.0±2.1 dB for diabetic patients and controls, respectively [p=0.962]. MS of paracentral area was 12.5±2.2 dB and 13.2±2.4 dB for diabetic patients and controls, respectively [p=0.251].”

We apologize for the misunderstanding about Figure 3. We changed the diabetic case of Figure 3 with a more representative case in line with the results of our study.

Furthermore, we included the possibility that more peripheral retina changes (outside 6 mm diameter) could be the first alterations in diabetic patients in the revised version of the manuscript (page 10, line 802-803): “Furthermore, our study analyzed mainly the 6x6 mm macular area; however, more peripheral retina analysis (outside 6 mm diameter) could show earlier alterations in diabetic patients”. However, we need to keep in mind that, as well shown by Rosenfeld’s and Wang’s groups, the resolution of OCT-A of scans greater than 6x6 mm is not enough to perform a reliable analysis of the perfusion density. For this reason, we performed only a 6x6 analysis and not a 12x12 or greater analysis.

Presentation/organization is confusing. It is unclear by the way that the file is formatted where the figure legends start and finish. In any case, this needs to be verified and clarified to make it so that the reader can follow easily the figures even independently from the result section (i.e. what does each letter in each figure mean?).

Thank you very much. We apologize for the confusing presentation, but the formatting of the paper was done by the journal after our submission. We revised the formatting of the file. We also clarified the figure legend as suggested.

Overall, the manuscript is fairly well-written but there remains a few instances of word misuse (disclose instead of found/observed) or poor formulation (“diabetic eyes revealed” instead of “the analysis of diabetic eyes revealed”) that need to be corrected.

Thank you very much. We corrected all the instances as suggested. We also performed a new spell-check of the manuscript.

Reviewer 2 Report

Riccardo Sacconi et al., assessed the vascular and neurodegenerative retinal alterations in type 1 diabetic patients without fundoscopic signs of diabetic retinopathy using Multimodal imaging. In this studies, they used structural OCT, OCT-A, DVA and microperimetry in a post-pediatric group of T1DM patients without fundoscopic signs of DR. Although study has some limitation as maintained but I have some concern need to be addressed.

comments:

Results section need more elaborative explanation of finding to understand better for the readers.

Discussion section need to be improved. Using this technology, Author did not find any change in retinal neurodegenerative alterations. It need more explanation and Limitation of these technology need be addressed properly.

Minor

Spelling mistake need to take care.

Author Response

Riccardo Sacconi et al., assessed the vascular and neurodegenerative retinal alterations in type 1 diabetic patients without fundoscopic signs of diabetic retinopathy using Multimodal imaging. In this studies, they used structural OCT, OCT-A, DVA and microperimetry in a post-pediatric group of T1DM patients without fundoscopic signs of DR. Although study has some limitation as maintained but I have some concern need to be addressed.

comments:

Results section need more elaborative explanation of finding to understand better for the readers.

Thank you very much for your suggestion. In the revised version of the manuscript, we improved each paragraph with a more elaborative explanation of the technology/results in order to better understand the results for the reader. Please see the revised version of the results’ section.

Discussion section need to be improved. Using this technology, Author did not find any change in retinal neurodegenerative alterations. It need more explanation and Limitation of these technology need be addressed properly.

Thank you very much for your suggestion. We better explained this point in the revised version of the manuscript (page 9, line 752-757): “Nevertheless, we cannot completely exclude the possibility that early photoreceptors changes, not detectable using structural OCT technology, could be the primary trigger in the DR development. Indeed, we tried to disclosed neurodegenerative and photoreceptor changes analyzing anatomical changes on structural OCT scans. However, this technology is not able to detect intracellular anatomical changes (not affecting the retinal thickness) and/or functional cellular changes.” and (page 10, line 797-801): “Indeed, structural OCT disclosed neurodegenerative changes analyzing the thickness of retinal layers. However, this technology is not able to detect intracellular anatomical changes (not affecting the retinal thickness) and/or functional cellular Another limitation of our study is that we performed the microperimetry in order to evaluate the retinal sensitivity, but other tests for retinal and visual function (i.e. visual field or dark adaptation examinations) could show different results”.

Spelling mistake need to take care.

Thank you for the suggestion. We performed a new spell-check of the document.

Reviewer 3 Report

The manuscript by Sacconi et al. examined and compared the retinas of young Type 1 diabetic (T1D) patients without diabetic retinopathy (DR) with the age-appropriate healthy controls.  They used morphological (OCT) and physiological (flicker light responses) parameters and found that T1D patients have decreased vascular densities in the deep capillary plexus (DCP).  Other vascular and neural retinal parameters are similar between T1D patients and healthy controls.

The authors suggested that early vascular dysfunction could be the primary trigger in the DR development, even though the authors did not exclude the neurodegenerative alternations in the development of DR.  However, the measurements from OCT and discussion on neurodegenerative changes are only in ganglion cells and nerves.  Please consider the possibility that early photoreceptors changes could be the primary trigger in the DR development.  This is based on: 1. Diabetic patients with retinitis pigmentosa rarely develop DR (Br J Ophthalmol 2001;85:366; American journal of ophthalmology 1984;97:788).  2. “Color vision” and photoreceptor electroretinography might have changed early on (Ophthalmic Physiol Opt 2010;30:717; Ophthalmic Physiol Opt 2010;30:705; Invest Ophthalmol Vis Sci 2012;53:741).  3. In animal studies by Tim Kern’s group showed that dysfunctional photoreceptors (e.g. under oxidative stress) lead to microvascular changes.  As the authors showed that DCP is different between T1D patients and healthy controls, it is possible that Tim Kern’s research could be correct.  

Since the authors have all the OCT images, please include the analyses in photoreceptor receptor layers (outer/inner segments, outer nuclear, outer plexiform layers) between T1D patients and healthy controls.  Please incorporate the above into Discussion.

Author Response

The manuscript by Sacconi et al. examined and compared the retinas of young Type 1 diabetic (T1D) patients without diabetic retinopathy (DR) with the age-appropriate healthy controls.  They used morphological (OCT) and physiological (flicker light responses) parameters and found that T1D patients have decreased vascular densities in the deep capillary plexus (DCP).  Other vascular and neural retinal parameters are similar between T1D patients and healthy controls.

The authors suggested that early vascular dysfunction could be the primary trigger in the DR development, even though the authors did not exclude the neurodegenerative alternations in the development of DR.  However, the measurements from OCT and discussion on neurodegenerative changes are only in ganglion cells and nerves.  Please consider the possibility that early photoreceptors changes could be the primary trigger in the DR development.  This is based on: 1. Diabetic patients with retinitis pigmentosa rarely develop DR (Br J Ophthalmol 2001;85:366; American journal of ophthalmology 1984;97:788).  2. “Color vision” and photoreceptor electroretinography might have changed early on (Ophthalmic Physiol Opt 2010;30:717; Ophthalmic Physiol Opt 2010;30:705; Invest Ophthalmol Vis Sci 2012;53:741).  3. In animal studies by Tim Kern’s group showed that dysfunctional photoreceptors (e.g. under oxidative stress) lead to microvascular changes.  As the authors showed that DCP is different between T1D patients and healthy controls, it is possible that Tim Kern’s research could be correct.

Thank you for your suggestion. We analyzed the OPL, ONL, EZ and OS surface thickness using structural OCT and we included these new analyses in the revised version of the manuscript (see response below). Although our analysis did not show any significant changes in the OPL, ONL, EZ, and OS thickness, we included this very interesting theory in the revised version of the manuscript (page 9, line 748-754): “Recent studies in animal models suggested that retinal photoreceptor cells have a crucial role in the pathogenesis of retinal microvascular lesion in diabetes [33]. However, our in vivo analysis did not show any significant changes in foveal OPL, ONL, EZ and OS surface thickness using structural OCT but a reduced PD of the DCP using OCT-A. Nevertheless, we cannot completely exclude the possibility that early photoreceptors changes, not detectable using structural OCT technology, could be the primary trigger in the DR development.”

Since the authors have all the OCT images, please include the analyses in photoreceptor receptor layers (outer/inner segments, outer nuclear, outer plexiform layers) between T1D patients and healthy controls.  Please incorporate the above into Discussion.

Thank you for your suggestion. We included the new analyses in photoreceptor layers (outer plexiform layer, outer nuclear layer, ellipsoid zone and outer segment surface) in the revised version of the manuscript. Methods (page 3, line 105-112): “Automated macular and peripapillary retinal nerve fiber layer (RNFL), macular ganglion cell complex (GCC), outer plexiform layer (OPL), and outer nuclear layer (ONL) thickness were also recorded using the abovementioned inbuilt software. … Furthermore, as Spectralis OCT does not provide an automatic segmentation of the ellipsoid zone (EZ), outer segment (OS) surface, and choroid, we manually measured the subfoveal EZ thickness (EZ was defined as the reflective layer situated posterior to the weak-reflecting ONL and anterior to the strong-reflecting RPE), OS thickness (OS was defined as the weak-reflecting layer between the EZ and RPE) and choroid thickness (ChT; ChT was defined as the distance between Bruch’s membrane interface and the sclerochoroidal interface). All values were manually measured by two expert readers (RS and EB) and the mean value was used for statistical analysis.”

Results (page 5, line 248-256): “In order to evaluate structural inner and outer retinal changes, we analyzed the thickness of RNFL, GCC, OPL, and ONL in the 3-mm and 6-mm-diameter macular subfields. On macular RNFL, GCC, OPL and ONL thickness analysis, no significant differences were disclosed comparing each 3-mm-diameter subfield (C, S, I, N and T) of diabetic patients with the corresponding subfield of control subjects, whereas with regard to GCC 6-mm-diameter subfield diabetic patients revealed a significant increased thickness in different quadrants compared to controls (Table 2 and 3). Furthermore, we analyzed the subfoveal EZ and OS thickness and we did not disclose any significant difference between diabetic patients and controls (EZ: 20.5±2.5 µm and 21.1±2.8 µm, respectively [p=0.355]; OS: 34.8±3.7 µm and 34.2±3.0 µm, respectively [p=0.497]).

We also incorporated the above point in the discussion (please see the response of above point).

Round 2

Reviewer 3 Report

There is no further suggestion from this reviewer.